# Prethermalization in one-dimensional quantum many-body systems with confinement

Stefan Birnkammer [1,2] ✉, Alvise Bastianello [1,2] & Michael Knap [1,2]

Unconventional nonequilibrium phases with restricted correlation spreading and slow entanglement growth have been proposed to emerge in systems with confined excitations, calling their thermalization dynamics into question. Here, we show that in confined systems the thermalization dynamics after a quantum quench instead exhibits multiple stages with well separated time scales. As an example, we consider the confined Ising spin chain, in which domain walls in the ordered phase form bound states reminiscent of mesons. The system first relaxes towards a prethermal state, described by a Gibbs ensemble with conserved meson number. The prethermal state arises from rare events in which mesons are created in close vicinity, leading to an avalanche of scattering events. Only at much later times a true thermal equilibrium is achieved in which the meson number conservation is violated by a mechanism akin to the Schwinger effect. The discussed prethermalization dynamics is directly relevant to generic one-dimensional, many-body systems with confined excitations.

Nonequilibrium states of quantum many-body systems play an important role in various fields of physics, including cosmology and condensed matter. Of particular interest is the time evolution of interacting quantum many-body systems that are well isolated from their environment[1,2]. This research has been fueled by the progress in engineering coherent and interacting quantum many-body systems which made it possible to experimentally study unconventional relaxation dynamics. A recent interest is to explore phenomena from high-energy physics with synthetic quantum systems in a controlled way; for example lattice gauge theories have been realized[3–9] and phenomena akin to quark confinement have been explored[3,6,10–12], with great emphasis on the atypical nonequilibrium features of confined systems. Confinement strongly affects the relaxation dynamics of the system, leading to unconventional spreading of correlations and slow entanglement growth[13–15], with striking signatures in the energy spectrum reminiscent of quantum scars[16,17]. In spite of many efforts, a proper characterization of the full many-body dynamics and thermalization in confined systems remains elusive so far.

An archetypical model to study confinement phenomena in condensed-matter settings is the Ising model with both transverse and longitudinal magnetic fields[18–23]. In this model, domain walls—interpreted as quarks—are pairwise confined into mesons by a weak longitudinal field; see Fig. 1a. A key feature of the model is the long lifetime of mesons, ascribed to a strong suppression of the Schwinger mechanism[24–29], which creates new quarks from the energy stored in the confining force and viceversa. Hence, except for some fine-tuned regimes[30–32], mesons are stable excitations. Due to the approximate conservation of the meson number, various exotic dynamical phenomena have been proposed, including Wannier-Stark localization[33–35] and time crystals[36]. Even though the realization of these phenomena does not require particular fine tuning, they arise in a regime in which interactions between mesons are extremely unlikely. The few-meson scattering has been recently considered[30,31,37,38], but so far, apart from special limits[39], the full many-body dynamics of confined systems has not been addressed. Irrespective of these exciting effects, the Ising model with longitudinal and transverse fields is non-integrable[19] and features a Wigner-Dyson level statistics of the eigenenergies[40]. Hence,

[1]Department of Physics, Technical University of Munich, 85748 Garching, Germany. [2]Munich Center for Quantum Science and Technology (MCQST), Schellingstr. 4, D-80799 München, Germany. ✉e-mail: stefan.birnkammer@tum.de

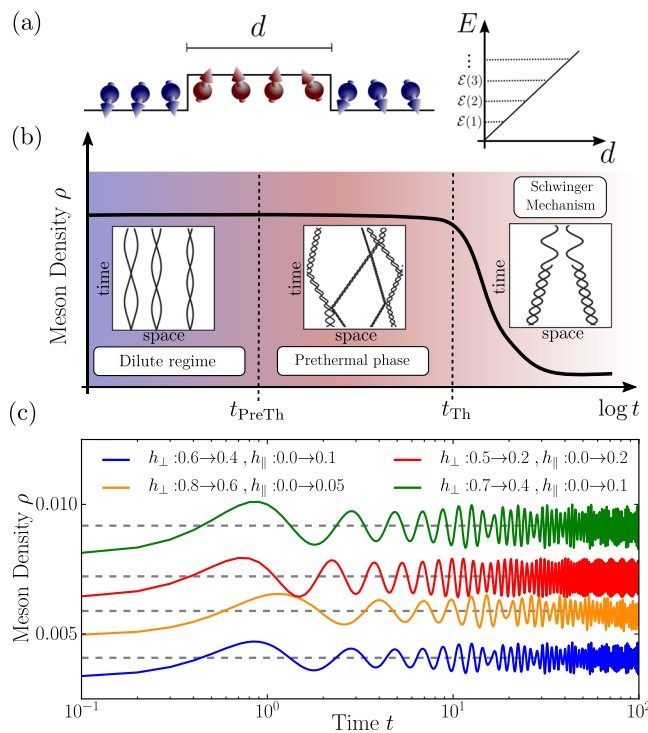

**Fig. 1 | Dynamics in the confined Ising chain. a** Pairs of domain walls, interpreted as mesons, are confined by the longitudinal field. **b** For weak quantum quenches of the transverse and longitudinal fields the Ising chain exhibits a multi-stage relaxation dynamics. Insets: typical domain wall trajectories in the different dynamical phases. At short times, $t < t_{\mathrm{PreTh}} \propto \rho^{-2} h_{\parallel}$ (with $\rho$ the density of mesons) a metastable state arises in which mesons are at rest and well separated. For intermediate times $t_{\mathrm{PreTh}} < t < t_{\mathrm{Th}}$, rare events initiate avalanches of scattering processes, leading to prethermal Gibbs ensemble with conserved density of mesons $\rho$. At late times $t > t_{\mathrm{Th}} \propto \exp[(\ldots)h_{\parallel}^{-1}]$, the Schwinger mechanism breaks the meson-number conservation leading to full thermalization. **c** The meson density $\rho$, computed with tensor network simulations, relaxes to the analytical prediction (dashed gray lines) of ref. [15] (see also supplementary information (Supplementary information for details on the confining dynamics; characterization of the prethermal state; initialization of moving mesons by staggered field pulses; further information on details of numerical simulations.)).

one would expect on general grounds[41–43] that the system thermalizes at late times and interactions between mesons can become relevant. Given this wealth of unconventional nonequilibrium phenomena and the discrepancy with the expected thermalization in non-integrable models, it is important to understand the mechanisms of relaxation and their timescales.

In this work, we investigate the relaxation dynamics of one-dimensional systems in the presence of confinement, with focus on the Ising chain as a primary example. Two scenarios could be envisioned for the thermalization process. The first one is that the Schwinger effect, leading to a violation of the meson-number conservation, could be the only responsible mechanism for equilibration, causing an extremely slow thermalization dynamics. A more exciting, second scenario involves an intermediate thermalization of the mesons themselves. Here, we show that indeed the second scenario is realized. Generic states first relax to a Gibbs ensemble in which the meson number is conserved up to extremely long times; Fig. 1b, c. We show that relaxation to this state is activated through rare events in which two mesons are produced in their vicinity, initiating an avalanche of scattering events. This prethermal state can then be understood as a dilute thermal gas of mesons with conserved meson density. Only at exponentially long times, the Schwinger mechanism causes a full thermalization of the system coupling sectors with a different number

of mesons. While we choose to focus on the Ising chain as the simplest example where both analytical and numerical progress can be made efficiently, our findings can be extended to generic confined many-body systems as we emphasize in the discussion section.

## Results

### Model and protocol

The Ising chain with both transverse and longitudinal fields is described by the Hamiltonian

$$\hat{H} = -\sum_j \left[ \hat{\sigma}_{j+1}^z \hat{\sigma}_j^z + h_\perp \hat{\sigma}_j^x + h_\parallel \hat{\sigma}_j^z \right]. \tag{1}$$

In the pure transverse-field regime ($h_\parallel = 0$) the model is equivalent to noninteracting fermions and exhibits spontaneous $\mathbb{Z}_2$−symmetry breaking for $|h_\perp| \leq 1$ in the thermodynamic limit. For $h_\perp \to 0$, the two degenerate ground states $|\mathrm{GS}_\pm\rangle$ are simple product states of maximally positive/negative magnetization, which are renormalized for finite transverse field, such that $\langle \mathrm{GS}_\pm| \hat{\sigma}_j^z |\mathrm{GS}_\pm\rangle = \pm\bar{\sigma}$, with $\bar{\sigma} = (1 - h_\perp^2)^{1/8}$ [44]. In this phase, the fermionic modes are interpreted as (dressed) domain walls (or kinks) relating the two vacua and are thus of topological nature. A small longitudinal field $h_\parallel > 0$ lifts the ground state degeneracy, leading to a low-energy "true vacuum" and a high-energy "false vacuum," and induces a pairwise linear potential $\propto 2h_\parallel \bar{\sigma}$ between kinks; Fig. 1a.

We consider the following quantum quench[13]: The system is initialized for $h_\parallel = 0$ in one of the two degenerate ground states (specifically, we select $\langle \hat{\sigma}^z \rangle > 0$) and then brought out of equilibrium by suddenly changing both the transverse and the longitudinal field components. Building on the knowledge of quenches in the transverse field only[45], one can argue that fermions are locally produced in pairs with opposite momenta[13], each of them having a dispersion $\epsilon(k) = 2\sqrt{(\cos k - h_\perp)^2 + \sin^2 k}$. However, pairs of fermions are then confined due to the finite longitudinal field $h_\parallel \neq 0$. For weak quenches, very few excitations are produced and, due to translational invariance, mesons are mostly initialized at rest and are well isolated. Their stability is guaranteed by the strong suppression of fermion number-changing processes. In the case of small transverse field ($|h_\perp| < 1/3$) two fermions cannot energetically couple to the four-fermion sector without using the energy stored in the false-vacuum string. Hence, this process resembles the false-vacuum decay, whose lifetime has been shown to scale exponentially with $h_\parallel^{-1}$ [24]. Even in the less restricted regime where the scattering of two fermions into four is energetically allowed ($1/3 < |h_\perp| < 1$), the cross section is induced by the weak longitudinal term, leading to a meson lifetime that scales algebraically in the longitudinal field $h_\parallel^{-3}$ [46]. To confirm this expectation, we perform tensor network simulations[47–49] based on the TenPy library[49] of the quantum quench and compute the meson density $\rho$; Fig. 1c. We checked convergence of our data with bond dimension on the shown timescales (data are shown for $\chi = 256$). In the limit of small $h_\parallel$ the meson number is conserved on the numerically accessible timescales (see Methods).

### Excitation spectrum and thermodynamics

Assuming that the meson number is conserved, we now study the thermodynamics of a gas of mesons, which is expected to describe the prethermal state. In the dilute regime, the mean-free path is much larger than the typical meson length. In a first approximation, we therefore neglect the effects that the size of the meson has on the thermodynamics. A convenient starting point is the semiclassical limit of a single meson, in which one treats the two fermions as point-like particles with coordinates $(x_{1,2}, k_{1,2})$ governed by the classical Hamiltonian

$$\mathcal{H} = \epsilon(k_1) + \epsilon(k_2) + 2h_\parallel \bar{\sigma}|x_1 - x_2|. \tag{2}$$

The semiclassical approximation holds when interactions cannot resolve the discreteness of the underlying lattice, i.e., for $h_{\parallel} \ll 1$. Hence, the position of the particle $x_{1,2}$ is a continuous variable. In the reduced two-body problem, the total momentum $k = k_1 + k_2$ of a meson is conserved, thus the dynamics of the relative coordinates $(q = (k_1 - k_2)/2, x = x_1 - x_2)$ is governed by $\mathcal{H}_{\rm rel}(q,x) = \epsilon(k/2 + q) + \epsilon(k/2 - q) + 2\bar{\sigma} h_{\parallel} |x|$. Then, the thermal probability of having a meson with a given energy and momentum is $P(E,k) = e^{-\beta(E-\mu)} \int \frac{dq\,dx}{(2\pi)^2} \delta(\mathcal{H}_{\rm rel}(q,x) - E)$, where the inverse temperature $\beta$ and chemical potential $\mu$ must be fixed by matching the initial average energy and meson density, respectively. The integral over the relative coordinates is most conveniently tackled by transforming to action-angle variables $(J, \phi)$[50], where $J \equiv \oint_{\mathcal{H}_{\rm rel}(q,x) = \mathcal{E}(J,k)} q\,dx$ labels the phase-space orbits of the classical motion and $\phi$ is a periodic variable $\phi \in [0,1]$, leading to $P(E,k) = e^{-\beta(E-\mu)} \int \frac{dJ}{(2\pi)^2} \delta(\mathcal{E}(J,k) - E)$. Leaving the classical limit, the energy levels become quantized according to the Bohr-Sommerfeld rule $J = 2\pi(n - 1/2)$, where $n$ is a natural number[20].

Away from the dilute regime mesons should be treated as extended objects and their thermodynamics needs to be suitably modified. To this end, we consider mesons as hard-rods of fixed length $\ell(J,k)$, the latter being the meson length averaged over one oscillation period. Within this assumption, $P(E,k)$ gets modified as

$$\frac{P(E,k)}{1 - \rho M} = e^{-\beta(E-\mu)} \int \frac{dJ}{(2\pi)^2} \delta(\mathcal{E}(J,k) - E) e^{-\rho \ell(J,k)(1 - \rho M)^{-1}} \quad (3)$$

with $\rho$ the meson density and $M$ the average meson length, which are self-consistently determined by $P(E,k)$; see also supplementary information (Supplementary information for details on the confining dynamics; characterization of the prethermal state; initialization of moving mesons by staggered field pulses; further information on details of numerical simulations.). The meson coverage $\rho M$ is connected to the magnetization of the Ising chain as $\rho M = 1/2 - \bar{\sigma}^{-1} \langle S^z \rangle$.

While we chose to present the thermodynamics from the semiclassical perspective for the sake of clarity, quantum effects can be important when the fermion bandwidth becomes comparable with the longitudinal field and the Born-Sommerfeld quantization is a poor approximation. In this regime, the classical Hamiltonian (2) can be directly promoted to a quantum object and explicitly diagonalized[15], thus replacing the $J-$ integration in Eq. (3) with a discrete sum (Supplementary information for details on the confining dynamics; characterization of the prethermal state; initialization of moving mesons by staggered field pulses; further information on details of numerical simulations.).

### Prethermalization of quantum mesons
In order to show that meson-meson scattering leads to a prethermal Gibbs Ensemble, we numerically calculate the time evolution in the subspace with a fixed number of mesons using exact diagonalization (see Methods). We consider a chain of length $L$ with periodic boundary conditions, and focus on the limit $0 < h_{\perp} \ll 1$ where fermions can be identified with domain walls. In this regime, $\bar{\sigma} \to 1$ and the confinement strength is determined by $h_{\parallel}/h_{\perp}$. We initialize the state in the form of moving wave packets and probe relaxation by tracking the meson momentum distribution; Fig. 2. Whereas for two mesons, energy and momentum conservation inhibits thermalization, see supplementary information (Supplementary information for details on the confining dynamics; characterization of the prethermal state; initialization of moving mesons by staggered field pulses; further information on details of numerical simulations.), for three mesons we observe the relaxation to the prethermal Gibbs ensemble; Eq. (3). Two-body scattering processes between different

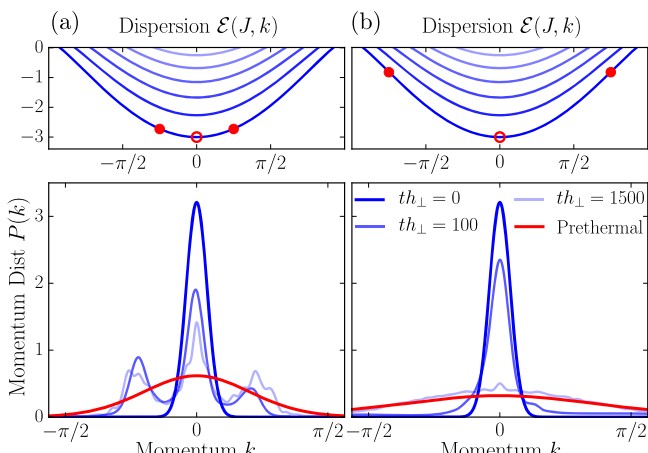

**Fig. 2 | Prethermalization of quantum mesons.** We create three mesons with Gaussian wave packets tuned to target the lowest energy band at momenta $\{-k_0, 0, k_0\}$ in a chain of length $L = 100$ with confinement field $h_{\parallel}/h_{\perp} = 0.1$ (see Methods). Energy bands are computed through a numerical solution of the two-fermions quantum Hamiltonian (Supplementary information for details on the confining dynamics; characterization of the prethermal state; initialization of moving mesons by staggered field pulses; further information on details of numerical simulations.). The evolution of the momentum distribution $P(k)$ of the meson initialized at rest (empty circle in upper panels) is shown. **a** For $k_0 = \pi/4$ the energy of the initial wave packets is below the second band of the single meson dispersion (upper panel), which causes meson scattering events to be elastic and prohibits relaxation to a prethermal ensemble (red curve). **b** For $k_0 = 3\pi/4$, the initial wave packets are resonant with other bands, which can then be populated, and lead to inelastic scattering. This quantum state relaxes to a prethermal configuration at late times. Prethermal curves are computed according to the hard-rods thermodynamics (3) using the exact quantum eigenfunctions rather than the semiclassical prediction (Supplementary information for details on the confining dynamics; characterization of the prethermal state; initialization of moving mesons by staggered field pulses; further information on details of numerical simulations.).

energy bands are responsible for the thermalization; Fig. 2b. For wave packets which are initialized with energies below the second band thermalization is largely suppressed, as two-body collisions become elastic due to momentum-energy conservation and three-body scattering events are unlikely; Fig. 2a. We provide additional details on the thermalization in the supplementary information (Supplementary information for details on the confining dynamics; characterization of the prethermal state; initialization of moving mesons by staggered field pulses; further information on details of numerical simulations.).

### Prethermalization through rare events
Equipped with the meson conservation, the thermodynamics of the prethermal state, and the quantum thermalization of a few mesons, we now study the full quench protocol. In order to access large system sizes and timescales, we use the Truncated Wigner Approximation[51] on the quantum dynamics projected in the fermion number conserving sector (see Methods). In order to study the relaxation dynamics, a precise knowledge of the excitation content of the initial state is crucial. The quantum quench of both the longitudinal and the transverse field excites dilute pairs of fermions with opposite momenta $(k, -k)$ at density $n(k)$, which can be computed from the quench parameters[13,15]. These pairs of fermions are then confined into mesons by the longitudinal field according to Eq. (2).

For small quenches within the ferromagnetic phase, the density of mesons is low. Typically, mesons are excited far apart and are thus isolated and at rest. In this scenario, inter-meson scattering and thermalization seems impossible. However, considering only the typical behavior is misleading, as the probability of creating two nearby

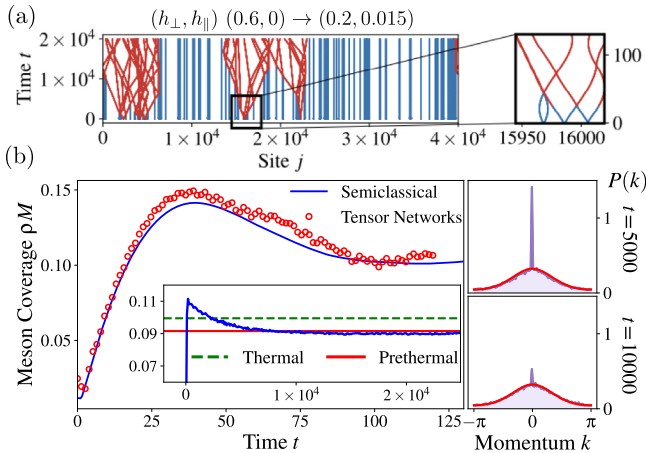

**Fig. 3 | Prethermalization through rare events. a** Typical semiclassical trajectories obtained with the Truncated Wigner Approximation. Most fermions belong to mesons at rest (blue lines), but rare events in which mesons are in close vicinity lead to an avalanche effect putting mesons in motion (red lines) and activating dynamics in the entire meson ensemble. **b** For comparably small values of $h_\parallel$ semiclassical results for the average meson coverage $\rho M$ agree well with exact quantum evolution obtained from tensor network techniques. (Inset) The semiclassical analysis reveals relaxation towards a prethermal plateau (red), which is distinct from the thermal state in the absence of meson conservation (green dashed). Side panels: Relaxation of the semiclassical ensemble is also reflected in the decay of the the momentum distribution $P(k)$ at $k \approx 0$. Thermal ($\mu = 0$) and prethermal ($\mu$ fixed by number of mesons) predictions are computed with Eq. (3), directly in the classical limit.

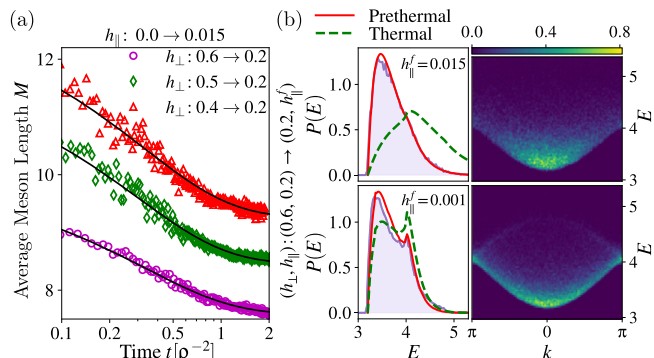

**Fig. 4 | Approaching the prethermal state and thermodynamics. a** The late-time relaxation of the average meson length $M$ to the prethermal plateau is well-described by the prediction $\langle \mathcal{O}(t) \rangle = \mathcal{O}_{\mathrm{PreTh}} + \Delta \mathcal{O} F(t v d \rho^2)$. The quantity $vd$ is obtained from a fit to the data. **b** The normalized one-meson phase-space occupation relaxes to a prethermal ensemble (prethermal: red continuous line; thermal: green dashed line; numerics: blue shaded area). Finite-density corrections are captured by the hard-rods approximation and cause an additional peak in the energy distribution $P(E)$ (bottom). The relative difference in the meson densities between the thermal and prethermal ensemble $\Delta \rho = (\rho_{\mathrm{PreTh}} - \rho_{\mathrm{Th}})/\rho_{\mathrm{Th}}$ are $\Delta \rho = 0.16$ and $\Delta \rho = \mathcal{O}(10^{-3})$ for $h_\parallel^f = 0.015$ (top) and $h_\parallel^f = 0.001$ (bottom), respectively. Thermal and prethermal observables are computed with Eq. (3), real-time evolution is obtained within the Truncated Wigner Approximation.

mesons is never strictly zero. To obtain a rough estimate, we consider the maximum size $d_{\max}$ that a meson can have when fermions are initially created at the same position, which is given by $d_{\max} = 4 h_\perp / (h_\parallel \bar{\sigma})$, and compare it with the meson density $\rho$. On a finite volume $L$, the probability $P(L)$ that $N = L\rho$ randomly distributed particles are placed at distance larger than $d_{\max}$ is $P(L) = \frac{1}{L^N} \prod_{j=0}^{N-1} (L - j d_{\max}) \simeq e^{-L\rho^2 d_{\max}/2}$. No matter how small the excitation density $\rho$ is, eventually in the thermodynamic limit the probability that all the excited mesons are far apart vanishes. Crucially, the rare nearby mesons scatter and acquire a finite velocity. These moving mesons consequently trigger an avalanche, that hits the surrounding mesons, and initiates prethermalization; see Fig. 3a for a typical meson configuration. In Fig. 3b, left, we first ensure that the semiclassical approximation is reliable for the chosen quench parameters, by comparing with tensor network simulations on the reachable timescale (convergence with bond dimension is checked; data shown for $\chi = 256$). Then, we use the semiclassical approach to probe extremely long times, observing prethermalization of both the meson coverage $\rho M$ (inset) and the momentum distribution of mesons $P(k)$ (right panels). For the latter the initial $\propto \delta(k)$ peak decays due to the aforementioned avalanche effect and relaxes to a smooth prethermal Gibbs distribution.

The density dependence of the prethermalization timescale $t_{\mathrm{PreTh}}$ can be understood as follows. Initially, the configuration consists of large regions of average size $\sim (\rho^2 d_{\max})^{-1}$ with mesons at rest separated by growing thermalizing domains. Hence, we estimate prethermalizing regions to cover the whole system on a typical time $t^* \sim (\rho^2 d_{\max} v)^{-1}$, where v is a typical velocity. Once all mesons are set in motion, two-body inelastic scatterings drive the relaxation of the system on a timescale $t^{**} \sim (\rho v)^{-1}$. At low excitation density, $t^* \gg t^{**}$, hence $t_{\mathrm{PreTh}} \sim (\rho^2 d_{\max} v)^{-1}$.

Building on this approximation, we can understand the relaxation of local observables by assuming that prethermalizing regions contribute with $\mathcal{O}_{\mathrm{PreTh}}$, while regions with static mesons retain the initial value $\mathcal{O}_0$ (after a short dephasing time). Hence, $\langle \mathcal{O}(t) \rangle$ follows the

average growth of thermalizing regions

$$\langle \mathcal{O}(t) \rangle = \mathcal{O}_{\mathrm{PreTh}} + \Delta \mathcal{O} \int_0^\infty \mathrm{d}D\, \mathcal{P}(D) \frac{D - 2vt}{D} \theta(D - 2vt), \qquad (4)$$

where we approximate each prethermalizing region to growth in a lightcone fashion with velocity v and $\Delta \mathcal{O} = \mathcal{O}_0 - \mathcal{O}_{\mathrm{PreTh}}$. Above, $\theta(x)$ is the Heaviside theta function $\theta(x > 0) = 1$ and zero otherwise, $D$ is the distance between two rare events which is distributed with probability distribution $\mathcal{P}(D)$. By its very definition, $\mathcal{O}_0$ can be computed as the late-time limit of the single meson approximation, since outside of the scrambling region the mesons are not interacting. Finally, the term $\frac{D - 2vt}{D}$ is nothing else than the portion of frozen region that remained after the thermalizing region propagated with velocity v inside of it. The last step is now to estimate $\mathcal{P}$. We have already computed the probability that, within a system of size $L$, there are no rare events. In the computation, we used the maximum extension of a meson $d_{\max}$ as an upper bound, but a better estimate is obtained using the average size of the excited mesons, which we call $d$. Hence, the probability that within an interval $L$ there are no rare events is $P(L) = e^{-L\rho^2 d/2}$. The distributions $P(L)$ and $\mathcal{P}(D)$ are related by $P(L) = \int_L^\infty \mathrm{d}D\, \mathcal{P}(D)$, leading to $\mathcal{P}(D) = \frac{\rho^2 d}{2} e^{-D\rho^2 d/2}$. With this approximation, Eq. (4) can be recast in a scaling form $\langle \mathcal{O}(t) \rangle = \mathcal{O}_{\mathrm{PreTh}} + \Delta \mathcal{O} F(t v \rho^2 d)$ with $F(\tau) = \int_\tau^\infty \mathrm{d}s\, e^{-s}(1 - \tau/s)$.

The full numerical results agree with this picture; Fig. 4a. Since $d_{\max} \propto h_\parallel^{-1}$, smaller longitudinal fields leads to a shorter prethermalization timescale for the same meson density $\rho$. Even in the less favorable case where $h_\perp$ is kept constant and only $h_\parallel$ is quenched (i.e., only the small longitudinal field is ultimately responsible of creating fermionic excitations), we find $\rho^2 \propto h_\parallel^{-2}$ (Supplementary information for details on the confining dynamics; characterization of the prethermal state; initialization of moving mesons by staggered field pulses; further information on details of numerical simulations.). Hence, there is in any case a separation of scales between the prethermalization time $t_{\mathrm{PreTh}} \propto h_\parallel^{-1}$ and the violation of meson-number conservation $t_{\mathrm{Th}} \sim \exp[(\ldots)h_\parallel^{-1}]$, consistently ensuring the existence of the prethermal regime for a large class of quenches.

In Fig. 4b we study the semiclassical prethermal regime for different confining strengths, but the same average density and energy. For $h_\parallel^f = 0.015$ (top) the average meson length is shorter than for

$h_\parallel^f = 0.001$ (bottom). We observe that the larger size of the mesons influences the phase-space distribution as follows: (i) it introduces a momentum-dependent cutoff in the energy, which is ultimately caused by the fact that the average meson length is bounded by the mean-free path, and (ii) the probability distribution is squeezed to the boundaries of the allowed phase space. A consequence of this is the emergence of a peak in the energy distribution corresponding to the Brillouin zone boundaries (compare bottom and top distibution functions). This effect is captured by our hard-rods approximation. For this choice of parameters, we observe the thermal number of mesons is lower than the prethermal one, hence thermalization is achieved by fusing small mesons into larger ones, i.e., by the reverse process of the Schwinger effect; Fig. 1b. In order to conserve the total mean energy, the thermal distribution has more high-energy mesons excited than the prethermal case. The difference between the prethermal and thermal state is reduced at higher meson densities, where the hard-rods correction penalizes large mesons.

By virtue of the simple underlying kinetic mechanism, the validity of our study is expected beyond the classical realm to hold in the quantum case as well, with an additional refinement. As previously mentioned, thermalization is activated by two-body scattering between different energy bands. Hence, the estimate of $t_{PreTh}$ should be corrected considering that only a fraction of $\rho$ is contributing to the inelastic scattering.

## Discussion

Confined spin chains exhibit an intriguing multi-stage thermalization dynamics. We show that not the Schwinger mechanism is responsible for activating transport, but rather rare events in which two mesons are generated in their vicinity lead to a prethermal regime, that can be understood as a thermal gas of mesons. The different mechanism ensures the separation of timescales and the existence of a prethermal regime. The prethermalization time can be greatly reduced by considering quench protocols that create mesons with non-zero velocity. This, for example, can be realized with spatially modulated pulses of the transverse field[52] (Supplementary information for details on the confining dynamics; characterization of the prethermal state; initialization of moving mesons by staggered field pulses; further information on details of numerical simulations.).

We used the Ising chain (1) as a prototypical model to demonstrate the rich relaxation dynamics. However, similar dynamics is expected in other confined many-body systems as well; for example lattice gauge theories[53–55]. Incidentally, we notice that the Ising chain (1) can be interpreted as a $\mathbb{Z}_2$−gauge theory in the zero charge sector, where matter degrees of freedom have been integrated out by virtue of the Gauss law[33]. A prominent example of a different lattice gauge theory is the $U(1)$ quantum link model

$$H_{QLM} = - \omega \sum_{j=1}^{L-1} (\phi_j^\dagger S_{j,j+1}^+ \phi_{j+1} + \text{h.c.})$$
$$+ \frac{m}{2} \sum_{j=1}^{L} (-1)^j \phi_j^\dagger \phi_j - 2h_\parallel \sum_{j=1}^{L-1} S_{j,j+1}^z, \quad (5)$$

where staggered Kogut-Susskind fermionic matter $\phi_j$[56] interacts via the gauge degrees of freedom encoded in the spin variables $S_{j,j+1}^\alpha$. In this model, (anti-)quarks correspond to defects in the staggered matter degrees of freedom and quark-antiquark pairs experience a linear confinement potential $\propto h_\parallel$. In a recent work[57], it has been understood that the Hamiltonian (5) maps to the Fendley-Sengupta-Sachdev Hamiltonian[58] describing one-dimensional Rydberg atom arrays[59] which may experimentally probe our findings. Within this implementation, the vacuum of the gauge theory is mapped into a chain where atoms are excited in their Rydberg state on even sites, then quark-antiquark pairs are excited by placing defects in this configuration.

Realizing a quantum quench akin to the one studied here, will thus lead to the same multi-stage thermalization dynamics. Further details can be found in supplementary information (Supplementary information for details on the confining dynamics; characterization of the prethermal state; initialization of moving mesons by staggered field pulses; further information on details of numerical simulations.).

Other experimentally relevant models with confinement can be realized in spin ladders[60–62] or long-range systems[11,25]. Particularly intriguing features can be expected for long-range models: in contrast to the short-range Ising chain (1) and the quantum link model (5) discussed above, long-range couplings induce slowly decaying (power-law) interactions between mesons which cannot be neglected. The long-range interactions can be envisaged to affect the approximation of dilute mesons, rendering prethermalization faster on the one hand, but making the approximation of the prethermal regime as a thermal gas of noninteracting mesons unreliable on the other hand. It would be interesting to extend our prethermal description to capture meson-meson interactions.

Another intriguing direction would be to address scenarios where the violation of the meson-number conservation is not negligible and must be properly considered. Can one observe and describe the drift to the thermal regime in such cases? A kinetic theory would require a quantitative understanding of meson creation-annihilation processes beyond the estimates discussed in this work.

## Methods

### Tensor network simulations

We used tensor network simulations to demonstrate the conservation of the meson number during the quantum evolution. Whereas time evolution can be carried out using the standard method of infinite Time-Evolving Block Decimation (iTEBD)[48,49], measurements of the meson number are more subtle. We outline how the mesonic number operator can be embedded efficiently in tensor network formalism.

The construction relies on the exact solution of the transverse Ising model, which we summarize in supplementary information (Supplementary information for details on the confining dynamics; characterization of the prethermal state; initialization of moving mesons by staggered field pulses; further information on details of numerical simulations.). Let $\{\gamma_k, \gamma_k^\dagger\}$ be the fermionic creation and annihilation operators that diagonalize the transverse-field Ising model in the absence of a longitudinal field $h_\parallel = 0$, the mesonic number operator is obtained as half of the mode number operator

$$2N_{mes} = \int_{-\pi}^{\pi} \frac{dk}{2\pi} \gamma_k^\dagger \gamma_k = \int_{-\pi}^{\pi} \frac{dk}{2\pi} \left( \cos^2\theta_k - \sin^2\theta_k \right) \alpha_k^\dagger \alpha_k +$$
$$+ \delta(0)\sin^2\theta_k + i\sin\theta_k \cos\theta_k \left( \alpha_{-k}\alpha_k - \alpha_k^\dagger \alpha_{-k}^\dagger \right), \quad (6)$$

where the $\hat{\alpha}_k = \cos\theta_k \hat{\gamma}_k + i\sin\theta_k \hat{\gamma}_{-k}^\dagger$ are the Jordan-Wigner fermions in the Fourier basis $\hat{c}_j = \int_{-\pi}^{\pi} \frac{dk}{\sqrt{2\pi}} e^{ikj} \hat{\alpha}_k$, which are eventually related to the original spin variables as $(\tilde{\sigma}_j^x + i\tilde{\sigma}_j^y)/2 = \exp\left( i\pi \sum_{i<j} \hat{c}_i^\dagger \hat{c}_i \right) \hat{c}_j^\dagger$. The Bogoliubov angle is tuned in such a way $\theta_k = -\frac{1}{2i}\log\left( \frac{h_\perp - e^{ik}}{(\cos k - h_\perp)^2 + \sin^2 k} \right)$.

The divergent factor $\delta(0)$ arises from equal-momentum commutation relation and it must be regularized $\delta(0) = L$, with the system size $L$. Moving to the coordinate space, the meson number can be thus written as

$$N_{mes} = \frac{1}{2} \sum_j \left\{ \sum_\ell \left[ f_1(\ell) c_{j+\ell}^\dagger c_j + \frac{1}{2} f_2(\ell) \left( c_{j+\ell} c_j + c_j^\dagger c_{j+\ell}^\dagger \right) \right] \right.$$
$$\left. + \int_{-\pi}^{\pi} \frac{dk}{2\pi} \sin^2\theta_k \right\}. \quad (7)$$

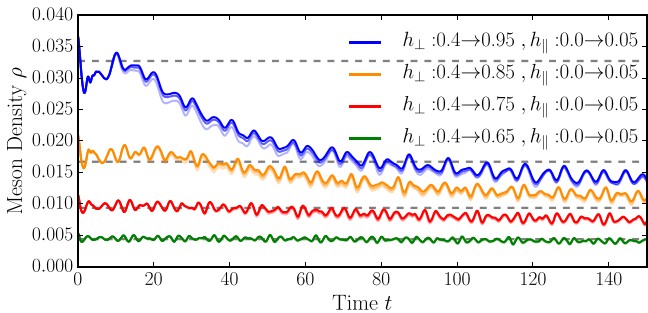

**Fig. 5 | Breakdown of meson-number conservation.** We show results for a quantum quench of the ground state with initial field configuration $(h_\perp, h_\parallel) = (0.4, 0.0)$ to values of the transverse field $h_\perp \in \{0.65, 0.75, 0.85, 0.95\}$ and additional confining longitudinal field $h_\parallel = 0.05$. For all quenches we show results obtained using iTEBD time evolution for a unit cell of $L = 40$ sites with bond dimensions of $\chi \in \{256, 384, 512\}$ (light to dark solid lines). We find that quenches close to the critical value of the transverse field only for a short time show the expected value of the meson number (gray dashed lines) before showing a decay in the number of mesons. The timescale, on which such a decay takes place increases and finally exceeds the numerically accessible times for quenches deep into the ferromagnetic phase ($h_\perp \ll 1$).

In Eq. (7) we introduced the functions $f_1(\ell), f_2(\ell)$ encoding the non-locality of the Jordan-Wigner mapping

$$f_1(\ell) = \int_{-\pi}^{\pi} \frac{dk}{2\pi} e^{ik\ell} \cos 2\theta_k \,, \quad f_2(\ell) = \int_{-\pi}^{\pi} \frac{dk}{2\pi} e^{ik\ell} \, i \sin 2\theta_k. \tag{8}$$

Finally, we can also invert the Jordan-Wigner mapping to obtain the expression in the spin basis $c_{j+\ell}^\dagger c_j = \sigma_{j+\ell}^- \left( \prod_{i=j}^{j+\ell-1} \sigma_i^z \right) \sigma_j^+$ and $c_{j+\ell} c_j = \sigma_{j+\ell}^+ \left( \prod_{i=j}^{j+\ell-1} \sigma_i^z \right) \sigma_j^+$.

Since $N_{\text{mes}}$ contains in general long-ranged terms an efficient representation in terms of an MPO strongly depends on the functional form of $f_1(\ell), f_2(\ell)$. For small values of the transverse field $h_\perp$ we find that both $f_1(\ell)$ and $f_2(\ell)$ can be approximated by an exponential decay for $l > 0$. This enables us to make use of the efficient representation of MPOs with coefficients exponentially decaying with distance discussed e.g., in ref. 48.

With this method, we can analyze quantum quenches in the Ising chain and follow the evolution of the number of mesons, checking whether it is approximately well-conserved or corrections are important. In Fig. 1c of the main text, we focus on parameter regimes where the meson number is oscillating around a constant value, in very good agreement with the analytic prediction of ref. 15. Oscillations have a technical origin and are due to the fact that, strictly speaking, it is not the number of fermions that is conserved, but rather the fermion number after a perturbatively small basis rotation[33]. Hence, in the original basis the fermion number couples to non-conserved quantities as well, which cause the small superimposed oscillations. To complement the analysis of Fig. 1c, in Fig. 5 we analyze quantum quenches where meson conservation is not a good approximation any longer. This can be achieved, for example, by tuning the post quench transverse field closer to the critical point, thus reducing the fermionic mass and enhancing the Schwinger mechanism. It is worth emphasizing that an efficient representation of $N_{\text{mes}}$ in terms of a MPO is no longer possible as $f_1(\ell), f_2(\ell)$ show deviations from an exponential decay for values of $h_\perp \to 1$. The meson number can, nonetheless, be computed by evaluating the terms contained in Eq. (7) individually and truncating the sum at large enough $\ell_{\text{max}}$. With this, we indeed observe that the difference between the numerical data and the analytic prediction grows with time as the post quench transverse field is tuned

sufficiently close to 1, in agreement with the observations of ref. 26. We, moreover, want to emphasize that results are converged with bond dimension $\chi$, as illustrated in Fig. 5. Smaller bond dimensions can lead to deviations of the time traces. Ensuring convergence of tensor network results for different choices of $\chi$ is hence crucial to estimate the actual relevance of meson-number-changing processes.

We notice the meson number decreases. Hence, rather than the usual Schwinger effect where a large meson decays in two (or more) smaller entities, what dominates the dynamics is the opposite process, namely inelastic scattering of two mesons that fuse and become a larger (i.e., more energetic) particle.

## Exact diagonalization within the few fermions sector

While tensor networks are numerically-exact methods, their applicability is constrained to short times by the entanglement growth, hence they cannot explore the prethermal regime. To overcome this restriction, we neglect the Schwinger mechanism and promote the number of mesons to an exact conservation law, thus projecting the dynamics within a sector with a fixed number of fermions. Furthermore, we wish to focus on the regime of a small transverse field where fermions are well approximated by domain walls. Hence, we consider the restricted Hilbert space $|j_1, j_2, \dots j_{2n-1}, j_{2n}\rangle = |\uparrow_1 \dots \uparrow_{j_1-1} \downarrow_{j_1} \dots \rangle$ generated by all the states with $n$ mesons, with ordered coordinates $j_{i+1} - j_i > 1$ and having values on the interval $[1, L]$. While the full Hilbert space in the spin basis grows as $2^L$, the restricted Hilbert space grows polynomially $\simeq \frac{1}{(2n)!} L^{2n}$ and much larger system sizes can be reached. This allows us to approach the regime where mesons are well separated, i.e., where our thermodynamic assumptions are valid. The same regime is naturally obtained after a quantum quench. By further taking into account translational invariance, the exponent of the polynomial growth in $L$ can be lowered by one unit, allowing us to simulate the dynamics of three mesons on $L = 100$ for very long times and eventually observing prethermalization (see Fig. 2). Further technical details on this method and benchmarks are discussed in supplementary information (Supplementary information for details on the confining dynamics; characterization of the prethermal state; initialization of moving mesons by staggered field pulses; further information on details of numerical simulations.).

## Semiclassical simulations

For large scale simulations in the semiclassical regime, we relied on a Truncated Wigner Approximation[51] which consists of the following steps (see also refs. 15, 38 for similar approximations)

(1)   Approximate the true Hamiltonian with the projected dynamics within the subspace with a fixed number of fermions (and its multiparticle generalization). This assumption is reliable as long as the Schwinger effect can be neglected.
(2)   Approximate the quantum evolution with a classical one:
   (a)   Replace quantum expectation values with proper averages over classical ensembles of particles.
   (b)   Replace the quantum evolution with properly chosen classical equations of motion, derived from the semiclassical Hamiltonian (2) (and its multiparticle generalization).
   (c)   Classical configurations are sampled from the classical statistical ensemble, then deterministically evolved with the equations of motion. The expectation values of observables is recovered by averaging over the initial conditions.

The Truncated Wigner Approximation is expected to work in the semiclassical regime, i.e., in the case of weak confinement; see below. However, as it is well known in the literature, one should be aware that this is an uncontrolled approximation in the sense that quantum corrections cannot be easily included in the approximation in a systematic way.

To further quantify the method and keep the notation under control, we now focus on the dynamics in the two-particle sector: the generalization to the multiparticle case can be directly obtained. Let us assume in full generality that the initial state is described by a density matrix $\hat{\rho}_{2pt}$, we focus on the matrix elements in a coordinate representation for the position of the two fermions $|j_1 j_2\rangle$. The Wigner distribution $W$ is defined through a partial Fourier transform of the matrix elements in the coordinate basis

$$\langle x_1 + y_1/2, x_2 + y_2/2|\hat{\rho}_{2pt}|x_1 - y_1/2, x_2 - y_2/2\rangle$$
$$= \int dk_1 dk_2 \, W(x_1, k_1, x_2, k_2) e^{iy_1 k_1 + iy_2 k_2}. \tag{9}$$

Above, one should impose integer values of the coordinates, but this restriction will not be important since classical physics emerges in the regime where the matrix elements are smooth functions of the coordinates, hence the discreteness of the lattice becomes irrelevant. We now move on to consider the dynamics by computing the Heisenberg equation of motion $i\partial_t \hat{\rho}_{2pt} = [\hat{H}_{2pt}, \hat{\rho}_{2pt}]$, where $\hat{H}_{2pt}$ is the quantum Hamiltonian projected in the two-fermions sector, namely the quantized version of Eq. (2). When expressing the Heisenberg equation of motion in terms of the Wigner distribution, one obtains after some straightforward calculations (we omit the $W$ − arguments for the sake of notation)

$$\partial_t W + [v(k_1)\partial_{x_1} + v(k_2)\partial_{x_2}]W$$
$$- V'(x_1 - x_2)\left(\partial_{k_1} - \partial_{k_2}\right)W \simeq 0 \tag{10}$$

where $v(k) = \partial_k \epsilon(k)$ and $V'(x) = \partial_x V(x)$ with $V(x) = 2h_\parallel \bar{\sigma}|x|$. The above equation is nothing else than the classical Liouville equation for the phase-space distribution $W(x_1, k_1, x_2, k_2)$ evolving with the classical Hamiltonian $\mathcal{H} = \epsilon(k_1) + \epsilon(k_2) + 2h_\parallel \bar{\sigma}|x_1 - x_2|$. In the derivation, one assumes that both the matrix element and the potential $V(x)$ are sufficiently smooth in the coordinates. Contributions neglected in the above equation are further orders in the derivative expansion. While $V(x)$ is not strictly speaking smooth, in the limit of weak longitudinal field the cusp in $V(x)$ gives negligible contributions. Notice that, if $h_\parallel$ is weak, smooth Wigner distributions will remain smooth during the evolution, ensuring the consistency of the approximation.

We finally turn to the problem of determining the initial Wigner distribution resulting from the quench protocol. To this end, we can resort to the quasiparticle picture of quantum quenches in the Ising chain[63], where the initial state is regarded as an incoherent gas of pairs of particles with opposite momentum $(k, -k)$, the probabilty distribution $n(k)$ of the pair can be computed from the exact solution of the quench in the transverse field[45] (see ref. 15 and supplementary information (Supplementary information for details on the confining dynamics; characterization of the prethermal state; initialization of moving mesons by staggered field pulses; further information on details of numerical simulations.) for details and corrections due to the finite longitudinal field). The distribution $n(k)$ fixes the probability distribution of a single pair of fermions: since pairs are independently created in a homogeneous fashion, we impose that pairs are distributed according to a Poisson distribution.

## Data availability
Data analysis is available on Zenodo upon reasonable request (all data are available upon reasonable request at https://doi.org/10.5281/zenodo.7034368).

## Code availability
Simulation codes are available on Zenodo upon reasonable request (simulation codes are available upon reasonable request at https://doi.org/10.5281/zenodo.7034368).

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

## Acknowledgements

We thank S. Scopa and P. Calabrese for collaboration on closely related topics and A. Lerose for useful discussions. We acknowledge support from the Deutsche Forschungsgemeinschaft (DFG, German Research Foundation) under Germany's Excellence Strategy–EXC–2111–390814868, TRR80 and DFG grants No. KN1254/1-2 and No. KN1254/2-1, the European Research Council (ERC) under the European Union's Horizon 2020 research and innovation programme (grant agreement No. 851161), as well as the Munich Quantum Valley, which is supported by the Bavarian state government with funds from the Hightech Agenda Bayern Plus.

## Author contributions

A.B. and S.B. carried out the numerical simulations contained in this work. M.K. supervised the work. All authors contributed critically to the writing of the manuscript and the interpretation of numerical and analytical results.

## Funding

## Competing interests

The authors declare no competing interests.
