## [Peer Review File · Nature Communications]

REVIEWER COMMENTS

Reviewer #1 (Remarks to the Author):

In the manuscript "Prethermalization in confined spin chains", the authors investigate the thermalization mechanism of a many-body spin system with analog confinement phenomena. In particular, the author focus on the mixed field Ising model that has been widely studied in recent years. The main result is that the thermalization is multi-stage, and in the first stage the system relax to a prethermal state described by Eq.(3) in the main text. Such a prethermal state conserves the meson number and is caused by inelastic scattering (rare events). The authors have shown several numerical evidence to support this argument, including prethermalization relaxation of few mesons (Fig.2) and multiple mesons by full quench (Fig.3 and Fig.4). On the technical level, the results are novel and the presentation is clear.

I have two major questions/concerns regarding the scope of manuscript.

i) The authors have argued that the relaxation phenomena is multistage. In the very long-time limit, the system will eventually reach to a true thermal state by Schwinger effect. However, there is no support to this argument. I understand the numerics would be challenging for large system size. It would be very helpful to even show small size numerics to support that there is indeed an additional stage of true thermalization.

ii) The authors have studied a very particular model in this manuscript, i.e. the transverse field Ising model with a longitudinal field. The authors do mention that there might be similar mechanism in long-range Ising model. It is not clear to me such a mechanism is general enough in other model, such as the lattice gauge theory with dynamical gauge field. If the presented mechanism is only limited to the particular model, I do not think it meets the acceptance criteria of Nature Communications.

Reviewer #2 (Remarks to the Author):

In this paper, the authors study a time scale of prethermalization in confined spin chains. Several time scales in the full many-body physics have been investigated, including dilution, prethermalization, and thermal equilibrium. The author uses theoretical arguments and numerical simulations to prove the validations of their theories.

The paper is clearly written and well-organized. However, I feel that several pieces are missing, making it hard to qualify NC, where we emphasize the originality and scope of the work. I would feel that it might be better to try other journals. Here is the reason:

1. The authors study confined Ising chains in 1+1 dimensions, using tensor network simulation methods. The system has been studied for a long time and has been investigated by numerous of people. However, it is hard to see the originality of their works and how to distinguish their results from others. For

instance, how generic the classification of those time scales is? From the first figure, it seems that it is not that well separated, especially the difference

between the dilution and prethermalization. In general, I feel it is a pure process of entanglement growth with particle production, and the authors just give names to something well-known.

2. It is not clear to me how to justify the validity and the error of the calculation. There is no fair or clear comparison of the exact diagonalization

methods, and also, there is no clear discussion on how much bond dimension it will need in theory. Those points might limit the accuracy of studies.

Thus, I do not think NC is that suitable for this work.

Reviewer #3 (Remarks to the Author):

Birnkammer et. al. find the existence of a prethermal regime in the confined mixed field Ising chain. The prethermal dynamics is described in terms of a semiclassical gas of mesons at low density, with conserved meson number. The mesons, which occur in several bands representing bound states of kinks of increasing separation, are shown to be accurately modeled as hard rods of increasing length (within this prethermal regime). This picture is checked with tensor network simulations of quantum quenches.

At parameterically long times in the parallel field, the prethermal regime should yield to a thermal regime -- but the timescales on which this happens are not studied explicitly in this paper. To my knowledge the existence of this prethermal regime is an original result, and it is certainly noteworthy. In addition to the discovery and thorough explanation of the result, there is a novel numerical trick for tracking the excitation number in tensor network numerics described in the supplemental material that is quite interesting.

The numerics explicitly show prethermalization and confirm explicit predictions that flow from their semiclassical analysis. The methodology is sound. Some greater detail should be supplied in a few places where the plots are not fully elucidated: in particular, how are the prethermal predictions marked by the gray lines in Fig 1c and the red lines in Figs 2, 3b, and 4b produced?

From the text, such lines might be produced through the semiclassical analysis or from an explicit quantum prethermal Hamiltonian. An additional detail that might be described is whether the prethermal Gibbs ensembles are described precisely by a prethermal Hamiltonian that is the full Hamiltonian projected into a fixed meson number sector, or whether there are higher order corrections to the prethermal Hamiltonian.

Additionally, one might describe how the mesons tracks shown in Fig 3a are produced.

Finally, I will say that the description of how thermalization might proceed in the paragraph describing the result of Fig 4b is a little hard to parse. Where it says "When the prethermal distribution is compared with the thermal expectation, more mesons with higher energy are thermally excited", I believe the figure shows that actually the thermal distribution has more higher energy mesons excited than the prethermal distribution. Is it clear that the higher energy mesons are also longer mesons, as described in the following sentence, rather than higher momentum?

Reply to Referee 1

In the manuscript “Prethermalization in confined spin chains”, the authors investigate the thermalization mechanism of a many-body spin system with analog confinement phenomena. In particular, the author focus on the mixed field Ising model that has been widely studied in recent years. The main result is that the thermalization is multi-stage, and in the first stage the system relax to a prethermal state described by Eq.(3) in the main text. Such a prethermal state conserves the meson number and is caused by inelastic scattering (rare events). The authors have shown several numerical evidence to support this argument, including prethermalization relaxation of few mesons (Fig.2) and multiple mesons by full quench (Fig.3 and Fig.4). On the technical level, the results are novel and the presentation is clear.

We would like to thank the Referee for the appreciation of our work [“*the results are novel and the presentation is clear*”]. The report points out two possible matters of concern, which we now address in detail in the response. We thank the Referee for inspiring us in further improving our work: we hope the resubmitted version will be regarded as suitable for publication in Nature Communications.

Below, we address in details the Report’s queries

- i) *The authors have argued that the relaxation phenomena is multistage. In the very long-time limit, the system will eventually reach to a true thermal state by Schwinger effect. However, there is no support to this argument. I understand the numerics would be challenging for large system size. It would be very helpful to even show small size numerics to support that there is indeed an additional stage of true thermalization.*

It is known from the literature that the longitudinal and transverse field Ising model which we consider is not integrable. This has been for instance demonstrated numerically in Ref. [1] by showing that the level statistics of the Ising chain with both longitudinal and transverse field is in fact of Wigner-Dyson type. Hence, the system is expected to obey the eigenstate thermalization hypothesis (ETH). As a consequence, the system should eventually thermalizes at very late times.

In order to observe the eventual thermalization in our system, we need to access a regime in which the mesons meson-number conservation is violated by inelastic collisions of two mesons that fuse in a single large meson (which happens at an extremely slow rate, due to the exponential suppression of the Schwinger mechanism). Hence, corrections on the meson number conservation take place at extremely long times.

Currently there exists no numerically exact technique that can capture the full evolution of the system. Tensor networks are not limited in system size, but strongly limited in the maximal entanglement of the many-body state. Since entanglement grows continuously in a large time-evolved many-body state, only intermediate time scales can be reached and

Figure R1: **Breakdown of meson number conservation.** We show results for a quantum quench of the ground state with initial field configuration $(h_{\perp}, h_{\parallel}) = (0.4, 0.0)$ to values of the transverse field $h_{\perp} \in \{0.65, 0.75, 0.85, 0.95\}$ and additional confining longitudinal field $h_{\parallel} = 0.05$. For all quenches we show results obtained using iTEBD time evolution for a unit cell of $L = 40$ sites with bond dimensions of $\chi \in \{256, 384, 512\}$. For quenches close to the critical value of the transverse field (blue), we find that the meson number (gray dashed lines) is conserved only for a short time before showing a decay. The timescale, on which such a decay takes place increases and finally exceed the numerically accessible times as one restricts quenches to stay deep in the ferromagnetic phase ($h_{\perp} \ll 1$).

the Schwinger regime cannot be accessed. Exact diagonalization on the other hand allows us to go to infinitely long times, but has severe restrictions in the accessible system size. This is why, we focus for this method on a conserved meson number which decreases the number of states and allows us to go to significantly larger systems. We emphasize that with small scale systems, relaxation dynamics might be strongly obscured due to finite size effects. This is why we cannot directly evaluate the dynamics of small systems and clearly see the discussed effects. Semiclassical techniques, such as the Truncated Wigner Approximation, do not capture the exact quantum dynamics, but are valuable methods for studying the qualitative behavior of the late time dynamics of correlated many-body systems, as research over the last decades has demonstrated in various examples. Due to these limitations of each of these methods, we require a combination of all of them to properly understand the complex many-body dynamics of our system.

Having said this, we fully agree with the Referee that it is important to show examples of regimes where the meson number is not conserved, even at the price of compressing the timescales and not being able to clearly tell apart the different stages in the thermalization process. To this end, we have performed new numerical simulations based on tensor networks. In particular, we approach the critical point $h_{\perp} \rightarrow 1$ (from below) where the Ising fermions become massless and the Schwinger mechanism is enhanced. Thus we have a chance of seeing the effect within the time scales accessible with tensor networks. In Fig. R1 we show results of tensor network evolution for quenches of the ground state according to a

choice for the magnetic fields of $(h_{\perp}, h_{\parallel}) = (0.4, 0.0)$. We perform different quenches in the transverse magnetic field $h_{\perp} \in \{0.65, 0.75, 0.85, 0.95\}$ while applying a fixed longitudinal field of $h_{\parallel} = 0.05$ after the quench. All simulations are carried using the iTEBD (infinte Time Evolving Block Decimation) algorithm for a unit cell of $L = 40$ sites, to evolve the iMPS initial state. In contrast to the results of Fig. 1 in the main text for this paramter regime we find a clear breakdown of meson number conservation on numerically accessible timescales for quenches close to the critical transverse field $h_{\perp}^c = 1.0$. Notice that the number of mesons is overall decreasing, hence fusion processes of two mesons merging in a larger entity dominate with respect to the opposite process.

We would like to point out that also insufficiently large bond dimension χ can lead to the impression of a decay in the meson number. In Fig.R1 we therefore check the convergence with three different values of the bond dimension $\chi \in \{256, 384, 512\}$ (light to dark solid lines) for each quench. The figure R1 has been added and properly discussed in our manuscript. As suggested by the referee this data shows the approach to the final thermal equilibrium in which the meson number conservation is violated.

- ii) *The authors have studied a very particular model in this manuscript, i.e. the transverse field Ising model with a longitudinal field. The authors do mention that there might be similar mechanism in long-range Ising model. It is not clear to me such a mechanism is general enough in other model, such as the lattice gauge theory with dynamical gauge field. If the presented mechanism is only limited to the particular model, I do not think it meets the acceptance criteria of Nature Communications.*

We chose to present our findings in the most celebrated toy model of confinement, i.e. the Ising chain. While we believe this choice enhances the clarity and readability of our work, we agree with the Referee that it can lead to the impression that the general mechanism we describe is relegated to a particular model, which is not true. In the revised version of the manuscript, we discuss how the same physics can be generically expected in one-dimensional gauge theories with confinement and connect with current experimental implementations. While the simplicity of the Ising model give access to many analytical results, the physics discussed in this spin chain can be easily carried over to many other examples with essentially no modifications. As such it will be important in various systems with confinement, which are currently tried to be realized for example with different platforms of synthetic quantum matter (ultracold atoms, Rydberg atoms, trapped ions, etc).

In particular, as an additional example we discuss the connection to a $U(1)$ quantum link model (QLM), which can be realized in current Rydberg atom quantum simulators [2]. The

model is described by the Hamiltonian

$$H_{\text{QLM}} = -\omega \sum_{j=1}^{L-1} (\phi_j^\dagger S_{j,j+1}^+ \phi_{j+1} + \text{h.c.}) + \frac{m}{2} \sum_{j=1}^L (-1)^j \phi_j^\dagger \phi_j - 2h_{\parallel} \sum_{j=1}^{L-1} S_{j,j+1}^z, \quad (1)$$

where staggered fermionic matter is described by the creators ϕ_j^\dagger and annihilators ϕ_j and we additionally include a gauge degree of freedom encoded by spin $\frac{1}{2}$ operators $S_{j,j+1}^z, S_{j,j+1}^+, S_{j,j+1}^-$ on the links $(j, j+1)$. The fermion content of the model in this staggered convention can be counted using a generalized number operator $n_j = \frac{1}{2}[1 - (-1)^j \phi_j^\dagger \phi_j]$. This relates to the picture that holes located at odd sites can be interpreted as antiquarks, whereas a particle located at an even site resembles a quark. Within the zero-gauge-charge sector $\left[S_{j,j+1}^z - S_{j-1,j}^z - \phi_j^\dagger \phi_j + \frac{1-(-1)^j}{2} \right] |\text{state}\rangle = 0$ the system features confinement, where h_{\parallel} plays the same role of the longitudinal field in the Ising chain while the hopping of topological excitations is determined by ω . A detailed description of this model and the connection to the discussed mechanism is provided in the revised version of our manuscript.

We are grateful to the Referee for their feedback that helped us in improving our manuscript, both in terms of readability and broad appeal. We think that the revised version now meets the criteria of Nature Communication. We hope the Referee shares our enthusiasm and can recommend the revised manuscript for publication.

Reply to Referee 2

In this paper, the authors study a time scale of prethermalization in confined spin chains. Several time scales in the full many-body physics have been investigated, including dilution, prethermalization, and thermal equilibrium. The author uses theoretical arguments and numerical simulations to prove the validations of their theories.

The paper is clearly written and well-organized. However, I feel that several pieces are missing, making it hard to qualify NC, where we emphasize the originality and scope of the work. I would feel that it might be better to try other journals. Here is the reason:

The authors study confined Ising chains in 1+1 dimensions, using tensor network simulation methods. The system has been studied for a long time and has been investigated by numerous of people. However, it is hard to see the originality of their works and how to distinguish their results from others.

We warmly welcome the appreciation of the Referee concerning the clarity of our manuscript. The Referee casts some doubts about the broad interest of our findings, but we believe these doubts can be discarded in view of our response below and of the many improvements made to our manuscript.

Before addressing the more technical questions, we would like to provide some general comments: we agree that the Ising model has been thoroughly studied in many papers, but the emphasis of our study is not on the model, but rather on some general pre-thermalization processes taking place in one-dimensional models with confinement. In this perspective, the Ising model is just a convenient example, well-known across many communities, that we choose due to its simplicity. This choice has been dictated to enhance the readability (as appreciated by the referee) of our manuscript and increase its appeal to a broad community, but the physics we discuss is not limited to this simple toy model. This point also connects with the report of the first Referee: in the revised manuscript we added a section to make explicit the broad applicability of our study and connecting with current experimental efforts. Let us also add that the confined Ising model has been successfully used as a paradigmatic toy model in many high-impact publications (e.g. [3, 4, 5] to mention just a few recent examples), without harming neither the generality of the findings or the appeal to a broad community. In particular, recent years have seen a deep interest in nonequilibrium dynamics of confined systems, see e.g. [3, 4, 6, 7, 8, 9, 10, 11, 5], but due to the difficulty of the problem these works address early time dynamics (what we call the dilute regime) or the few-body regime.

Our study stands out as the only one (up to our knowledge) that studies inherently many-body dynamics of confined systems.

We now address the two main points raised by the Referee

- i) *For instance, how generic the classification of those time scales is? From the first figure,*

it seems that it is not that well separated, especially the difference between the dilution and prethermalization. In general, I feel it is a pure process of entanglement growth with particle production, and the authors just give names to something well-known.

We would like to emphasize that the purpose of Fig. 1 is to highlight the emergence of a new conservation law, which is the number of mesons. In particular both in the dilute and in the prethermal regime the number of mesons are conserved, and only in the thermalizing regime the Schwinger effect leads to a violation of the meson conservation. Hence, the meson number shown in Fig. 1 is not the right quantity to distinguish the dilute and the prethermal regime.

A quantitative characterization of the different dynamical phases and their timescale requires a careful analysis of several observables, which we carry out in the main body of our manuscript. In particular, in Fig 1 (b) we relate the introduced terminology of dilute and prethermal regime to typical configurations found in our semiclassical simulations (insets). Whereas the dilute regime is dominated by independent oscillations of static mesons, the prethermal regime is characterized by collisions between moving mesons. As a consequence, observables in the dilute regime will be highly sensitive to the initial state preparation, while the emergent prethermal regime is completely characterized by the total energy and number of mesons. These processes and the associated time scales are elucidated, for example, in the evolution of the momentum and energy distribution (Fig 2 and Fig 4) and thoroughly discussed in the main text.

We would like to add that, as the Referee points out, prethermalization and finally thermalization is eventually a process of entanglement growth with possible particle production. However, it is an large and active area of research to understand the different dynamical regimes which arise (i.e., passing from the prethermal to the thermal phase), and in particular for dynamics induced in systems with confinement. As suggested in the pioneering paper [3] and many subsequent studies, prior to our work these systems were thought to not show any signature of (pre-)thermalization for exponentially long times in the confining strength. We provide a simple physical picture and extensive numerical studies to demonstrate the opposite. In order to clarify these points of confusion, we have in several places improved the clarity of our manuscript.

- ii) *It is not clear to me how to justify the validity and the error of the calculation. There is no fair or clear comparison of the exact diagonalization methods, and also, there is no clear discussion on how much bond dimension it will need in theory. Those points might limit the accuracy of studies.*

The Referee points out that the previous version of our manuscript did not contain a explicit comparison between exact diagonalization (ED) on the few-kinks subspace and exact simulations using tensor networks (TN). We have performed new simulations obtaining

Figure R2: **Comparison of exact diagonalization in few kink subspace and tensor networks.** We benchmark the evolution results for a N meson initial state $|\psi_i^{(N)}\rangle$ obtained using exact diagonalization (ED) in the few kink subspace against numerically exact results from tensor network (TN) evolution. We consider values for the transverse field of $h_\perp \in \{0.05, 0.075, 0.1\}$ and fixed longitudinal field $h_\parallel = 0.05$ while numbers of $N \in \{1, 2, 3\}$ mesons in the chain are tested. We find very good agreement between results of ED and TN in systems of $L = 60$ sites and bond dimensions up to $\chi = 256$ for small values of $h_\perp = 0.05$. Considering larger values of the transverse field $h_\perp = 0.1$ we find qualitative features of the TN evolution to be reproduced by ED results. The small offset is understood from the dressing of the fermionic domain walls at larger values of h_\perp .

such a comparison and include it in the revised version of our manuscript. We focus on different values of the magnetic fields (h_\perp, h_\parallel) and different meson numbers N . The initial state for the evolution is given by a homogenous superposition of N isolated spin flips applied to the ground state $|0\rangle$ in a system of $L = 60$ sites with open boundary conditions

$$|\psi_i^{(N)}\rangle = \sum_{\{j_1, j_2, \dots, j_N\}} \sigma_{j_1}^x \sigma_{j_2}^x \dots \sigma_{j_N}^x |0\rangle. \quad (2)$$

$\{1 < j_1 < j_2 < \dots < j_N < L \mid j_{n+1} - j_n > 1 \forall 1 \leq n < N\}$. For small values of the transverse field h_\perp , where fermionic domain walls are close to sharp kinks in the order parameter, this state has large overlap with a N meson configuration. Each meson thereby has the smallest possible extent of a single lattice spacing. The results for the evolution using ED and TN methods are shown in Fig. R2. We find very good agreement between both methods for all tested values of $N \in \{1, 2, 3\}$ and small values of $h_\perp = 0.05$. Moreover,

we find qualitative features of the evolution like the oscillation frequency of ρM preserved also for larger values of $h_{\perp} = 0.1$, however, we can identify an additional offset between results of TN respectively to predictions of ED. This deviation is expected when considering larger values of h_{\perp} , where the functional form of the domain wall gets dressed by the finite transverse field. As a consequence we expect to find corrections to the predictions from ED of sharp kink-like domain walls, as reflected in Fig. R2.

Moreover, we agree with the Referee that numerical parameters such as the value of the bond dimension χ for tensor network studies should be addressed in detail. In general all the reported results have been checked for convergence in bond dimension χ and plots show the largest tested value of χ . In addition we now discuss the effect of the bond dimension χ in the revised version of the manuscript, see e.g. Fig 5 on the breakdown of meson number conservation. In this figure we show different values for $\chi \in \{256, 384, 512\}$ and find convergence. Smaller bond dimensions would lead to deviations of the time traces.

We would like to thank the Referee for highlighting the benefits of such a benchmark between the numerical methods, which is now discussed in an additional section, and for highlighting possible points of confusion, which we have clarified in the revised manuscript. We hope that the revised version of our manuscript will be considered suitable for publication in Nature Communications.

Reply to Referee 3

Birnkammer et. al. find the existence of a prethermal regime in the confined mixed field Ising chain. The prethermal dynamics is described in terms of a semiclassical gas of mesons at low density, with conserved meson number. The mesons, which occur in several bands representing bound states of kinks of increasing separation, are shown to be accurately modeled as hard rods of increasing length (within this prethermal regime). This picture is checked with tensor network simulations of quantum quenches. At parameterically long times in the parallel field, the prethermal regime should yield to a thermal regime – but the timescales on which this happens are not studied explicitly in this paper. To my knowledge the existence of this prethermal regime is an original result, and it is certainly noteworthy. In addition to the discovery and thorough explanation of the result, there is a novel numerical trick for tracking the excitation number in tensor network numerics described in the supplemental material that is quite interesting.

We thank the Referee for their appreciation of our work [*“the existence of this prethermal regime is an original result, and it is certainly noteworthy”*] and [*“there is a novel numerical trick for tracking the excitation number in tensor network numerics described in the supplemental material that is quite interesting.”*].

We very much appreciate the careful reading of our work and for raising the points of potential obscurity. We now clarified these requests in the resubmitted version and below

- i) *The numerics explicitly show prethermalization and confirm explicit predictions that flow from their semiclassical analysis. The methodology is sound. Some greater detail should be supplied in a few places where the plots are not fully elucidated: in particular, how are the prethermal predictions marked by the gray lines in Fig 1c and the red lines in Figs 2, 3b, and 4b produced? From the text, such lines might be produced through the semiclassical analysis or from an explicit quantum prethermal Hamiltonian. An additional detail that might be described is whether the prethermal Gibbs ensembles are described precisely by a prethermal Hamiltonian that is the full Hamiltonian projected into a fixed meson number sector, or whether there are higher order corrections to the prethermal Hamiltonian.*

The analytically-computed reference values of Fig. 1 (c) for the meson number after a quantum quench are computed resorting to the methods of Ref. [6]. The result of this calculation is reported in the first section of the supplementary material. We have improved the caption of Fig. 1 to clarify this this point.

The prethermal expectation values shown in Figs 2, 3b and 4b are computed within the hard-rod thermodynamics described in the text and further explained in the supplementary material. In Fig 2, i.e. deep in the quantum regime, the exact quantum spectrum of the Hamiltonian projected in the few-kink subspace is considered, while semiclassical predictions are used in Figs 3b and 4b. We clarify these points in the captions of the figures in

the revised version of our manuscript. Meson energies and wavefunctions are computed by projecting the full Hamiltonian in the few fermions (or kinks for small h_{\perp}) sector without further contributions, but in the thermodynamics we include the hard-rods correction. We clarify the technicalities in the revised manuscript.

- ii) *Additionally, one might describe how the mesons tracks shown in Fig 3a are produced.*

The fermion trajectories shown in Fig. 3 (a) are obtained through the semiclassical Truncated Wigner Approach (TWA). Within this approach, one describes the quantum state as a statistical ensemble of classical excitations (the fermions) whose initial phase-space distribution is properly tuned to match the quantum result. In this case, it consists of a homogeneous distribution of locally-generated pairs of fermions with opposite momenta $(k, -k)$ and a momentum distribution $n(k)$ given by the quantum solution of the quench protocol. After initialization, the excitations are propagated classically with the proper equations of motion, in this case those of confined particles derived from Eq. (2) (and its multiparticle generalization).

In this picture, one drags particle configurations from the initial distribution, deterministically evolve with the equation of motion and finally average over the initial conditions to reconstruct the behavior of observables.

The trajectories of Fig. (3a) are the trajectories obtained from a typical initial configuration, the different coloring (blue and red) are a guide to the eye to distinguish fermions bound in a meson at rest (blue line) or forming a moving meson (red) which can thus activate transport. The color transition from blue to red highlights the transition from the dilute regime to the prethermal phase. We include additional details on TWA in the revised version of the manuscript and clarify how the tracks in Fig. 3a are computed.

- iii) *Finally, I will say that the description of how thermalization might proceed in the paragraph describing the result of Fig 4b is a little hard to parse. Where it says "When the prethermal distribution is compared with the thermal expectation, more mesons with higher energy are thermally excited", I believe the figure shows that actually the thermal distribution has more higher energy mesons excited than the prethermal distribution. Is it clear that the higher energy mesons are also longer mesons, as described in the following sentence, rather than higher momentum?*

We apologize for the confusion concerning the comparison between the prethermal and thermal meson distribution in connection with Fig. 4b. We indeed meant (but very poorly wrote) that [*"the thermal distribution has more higher energy mesons excited than the prethermal distribution"*]. We rephrased and clarified this passage.

For the choice of parameters of Fig. 4b, it turns out that the number of mesons of the thermal distribution is lower than that of the prethermal state. Hence, more energetic mesons are thermally excited in order to keep the average energy constant.

$$h_{\parallel} = 0.015, h_{\perp} = 0.2$$

Figure R3: **Average meson length** $\ell(E, K)$. We show the average meson length $\ell(E, K)$ obtained from averaging the distance between the two fermionic particles forming a single meson over the semiclassical oscillation period of the two particle problem. We find that mesons with higher energy typically also are associated with longer average lengths $\ell(E, K)$. The shown results correspond to a field configuration of $(h_{\perp}, h_{\parallel}) = (0.2, 0.015)$ also used in Fig. 4b of the main text.

Let us now comment on the connection between the meson length and their energy. In general, the Referee is right and the average length also depends on the meson momentum. Nonetheless, for large mesons where the potential energy stored in the string tension between the fermions dominates over the kinetic energy of the latter, the energy is directly proportional to the average meson length. Therefore, placing a cutoff on the maximum length of the meson is eventually translated in a cutoff in its energy, which is what we demonstrate in Fig. 4b; for a more detailed relation between the average meson length ℓ and its energy E and momentum K see Fig. R3.

We thank the referee for their time in reading our manuscript and for the many useful comments. The Referee's report greatly helped us in improving our work, which we hope can be now recommended for publication in Nature Communications.

References

- [1] Hyungwon Kim and David A. Huse. “Ballistic Spreading of Entanglement in a Diffusive Nonintegrable System”. In: *Phys. Rev. Lett.* 111 (12 2013), p. 127205. DOI: [10.1103/PhysRevLett.111.127205](https://doi.org/10.1103/PhysRevLett.111.127205). URL: <https://link.aps.org/doi/10.1103/PhysRevLett.111.127205>.
- [2] Federica M. Surace et al. “Lattice Gauge Theories and String Dynamics in Rydberg Atom Quantum Simulators”. In: *Phys. Rev. X* 10 (2 2020), p. 021041. DOI: [10.1103/PhysRevX.10.021041](https://doi.org/10.1103/PhysRevX.10.021041). URL: <https://link.aps.org/doi/10.1103/PhysRevX.10.021041>.
- [3] Marton Kormos et al. “Real-time confinement following a quantum quench to a non-integrable model”. In: *Nature Physics* 13.3 (2017), pp. 246–249. ISSN: 1745-2481. DOI: [10.1038/nphys3934](https://doi.org/10.1038/nphys3934). URL: <https://doi.org/10.1038/nphys3934>.
- [4] Olalla A. Castro-Alvaredo et al. “Entanglement Oscillations near a Quantum Critical Point”. In: *Phys. Rev. Lett.* 124 (23 2020), p. 230601. DOI: [10.1103/PhysRevLett.124.230601](https://doi.org/10.1103/PhysRevLett.124.230601). URL: <https://link.aps.org/doi/10.1103/PhysRevLett.124.230601>.
- [5] Andrew J. A. James, Robert M. Konik, and Neil J. Robinson. “Nonthermal States Arising from Confinement in One and Two Dimensions”. In: *Phys. Rev. Lett.* 122 (13 2019), p. 130603. DOI: [10.1103/PhysRevLett.122.130603](https://doi.org/10.1103/PhysRevLett.122.130603). URL: <https://link.aps.org/doi/10.1103/PhysRevLett.122.130603>.
- [6] Stefano Scopa, Pasquale Calabrese, and Alvisio Bastianello. “Entanglement dynamics in confining spin chains”. In: *Phys. Rev. B* 105 (12 2022), p. 125413. DOI: [10.1103/PhysRevB.105.125413](https://doi.org/10.1103/PhysRevB.105.125413). URL: <https://link.aps.org/doi/10.1103/PhysRevB.105.125413>.
- [7] Alessio Lerose et al. “Quasilocalized dynamics from confinement of quantum excitations”. In: *Phys. Rev. B* 102 (4 2020), p. 041118. DOI: [10.1103/PhysRevB.102.041118](https://doi.org/10.1103/PhysRevB.102.041118).
- [8] Paolo Pietro Mazza et al. “Suppression of transport in nondisordered quantum spin chains due to confined excitations”. In: *Phys. Rev. B* 99 (18 2019), p. 180302. DOI: [10.1103/PhysRevB.99.180302](https://doi.org/10.1103/PhysRevB.99.180302). URL: <https://link.aps.org/doi/10.1103/PhysRevB.99.180302>.
- [9] O. Pomponio et al. *Bloch oscillations and the lack of the decay of the false vacuum in a one-dimensional quantum spin chain*. 2021. arXiv: [2105.00014](https://arxiv.org/abs/2105.00014) [[cond-mat.stat-mech](https://arxiv.org/abs/2105.00014)].
- [10] Mario Collura et al. *Discrete time-crystalline response stabilized by domain-wall confinement*. 2021. arXiv: [2110.14705](https://arxiv.org/abs/2110.14705) [[cond-mat.stat-mech](https://arxiv.org/abs/2110.14705)].

- [11] Neil J. Robinson, Andrew J. A. James, and Robert M. Konik. “Signatures of rare states and thermalization in a theory with confinement”. In: *Phys. Rev. B* 99 (19 2019), p. 195108. DOI: [10.1103/PhysRevB.99.195108](https://doi.org/10.1103/PhysRevB.99.195108). URL: <https://link.aps.org/doi/10.1103/PhysRevB.99.195108>.

REVIEWERS' COMMENTS

Reviewer #2 (Remarks to the Author):

The authors answer my questions and I think now it is in a good shape and deserves publications.

Reviewer #3 (Remarks to the Author):

Birnkammer et. al. find the existence of a prethermal regime in the confined mixed field Ising chain. The result is novel and noteworthy. The prethermal dynamics is quantified numerically with tensor network simulations and effectively described in terms of a semiclassical gas of mesons at low density. The analysis verifies this picture with multiple lines of numerical evidence using sound methodology.

The revised manuscript addresses the points where this reviewer was confused upon reading the draft. The addition of new ED numerics support the soundness of their tensor network calculations in the regime that they study.

I recommend publication.